# A Review on Silver Nanoparticles: Classification, Various Methods of Synthesis, and Their Potential Roles in Biomedical Applications and Water Treatment

Muhammad Zahoor [1,*], Nausheen Nazir [1], Muhammad Iftikhar [1], Sumaira Naz [1], Ivar Zekker [2,*], Juris Burlakovs [3], Faheem Uddin [4], Abdul Waheed Kamran [5], Anna Kallistova [6], Nikolai Pimenov [6] and Farhat Ali Khan [7]

1 Department of Biochemistry, University of Malakand, Chakdara 18800, Pakistan; nausheen.nazir@uom.edu.pk (N.N.); miftekhar355@gmail.com (M.I.); sumaira.biochem@gmail.com (S.N.)
2 Faculty of Science, Institute of Chemistry, University of Tartu, 51014 Tartu, Estonia
3 Institute of Forestry and Rural Engineering, Estonian University of Life Sciences, 51006 Tartu, Estonia; Juris.burlakovs@emu.ee
4 Department of Electrical Engineering, University of Engineering & Technology, Mardan 23200, Pakistan; faheemproudpak@gmail.com
5 Department of Chemistry, University of Malakand, Chakdara 18800, Pakistan; waheedkamran1989@gmail.com
6 Research Centre of Biotechnology of the Russian Academy of Sciences, Winogradsky Institute of Microbiology, 119071 Moscow, Russia; kallistoanna@mail.ru (A.K.); npimenov@mail.ru (N.P.)
7 Department of Pharmacy, Shaheed Benazir Bhutto University, Sheringal 18050, Pakistan; farhatkhan2k9@yahoo.com
* Correspondence: mohammadzahoorus@yahoo.com (M.Z.); ivar.zekker@ut.ee (I.Z.)

**Abstract:** Recent developments in nanoscience have appreciably modified how diseases are prevented, diagnosed, and treated. Metal nanoparticles, specifically silver nanoparticles (AgNPs), are widely used in bioscience. From time to time, various synthetic methods for the synthesis of AgNPs are reported, i.e., physical, chemical, and photochemical ones. However, among these, most are expensive and not eco-friendly. The physicochemical parameters such as temperature, use of a dispersing agent, surfactant, and others greatly influence the quality and quantity of the synthesized NPs and ultimately affect the material's properties. Scientists worldwide are trying to synthesize NPs and are devising methods that are easy to apply, eco-friendly, and economical. Among such strategies is the biogenic method, where plants are used as the source of reducing and capping agents. In this review, we intend to debate different strategies of AgNP synthesis. Although, different preparation strategies are in use to synthesize AgNPs such as electron irradiation, optical device ablation, chemical reduction, organic procedures, and photochemical methods. However, biogenic processes are preferably used, as they are environment-friendly and economical. The review covers a comprehensive discussion on the biological activities of AgNPs, such as antimicrobial, anticancer anti-inflammatory, and anti-angiogenic potentials of AgNPs. The use of AgNPs in water treatment and disinfection has also been discussed in detail.

**Keywords:** silver nanoparticles; physical methods; chemical methods; green synthesis; classification; AgNPs in water treatment

## 1. Introduction

Nanotechnology has been considered one of the essential fields in current science that permits scientists to acquire remarkable nanoparticle (NP) size innovations. By definition, particles having diameters below 100 nm are referred to as NPs. Different strategies are used to prepare metal NPs with the smallest possible size. Broadly, the methods may be categorized into chemical, physical, photochemical, and organic techniques [1]. The

synthesis of NPs from several metals, such as palladium, tin, copper, silver, and gold, has acquired greater interest due to their specific properties and application in different fields. Over the period, various synthetic techniques for the synthesis of NPs are constantly reported. However, most of the available strategies are expensive and not eco-friendly. Apart from the type of synthetic method used, different factors, such as temperature, dispersing agent, and surfactant, substantially impact the quality and quantity of the synthesized NPs in the long run. However, to obtain NPs with a particular shape, size, and better properties, scientists are looking for an eco-friendly route along with economic benefits. Considering the adverse outcomes of different techniques on living organisms, scientists prefer biogenic approaches for NP synthesis because they can produce NPs on a large scale with less or no risks to the environment [2]. In the biogenic approach of the synthesis, bacteria, fungi, algae, and plant extracts are used as a reductant for the fabrication of metal NPs [3].

Silver NPs (AgNPs) are extensively used in diverse fields, such as medicine, food, healthcare, and industrial purposes, because of their particular physical and chemical properties, morphology and distribution, size, shape, and high surface area. They have exhibited better applications in optical, electrical, and thermal devices with high electrical and heat conductivity and a wide area of organic chemistry-related applications. They are employed in medical device coatings, optical sensors, cosmetics, different pharmaceutical industry products, and the food industry. Their use in diagnostics, theranostics, and drug delivery as antimicrobial, anti-inflammatory, and anticancer agents is also worth mentioning [2,4,5]. They are utilized in electronics, mechanics, environment, biomedical, drug and gene delivery, catalysis, optics, space industries, energy science, chemical industries, single-electron transistors, light emitters, non-linear optical devices, and other applications. Over the past two decades, the field has flourished by manufacturing nano-metric size substances, creating a new subject of study because of AgNPs' versatile applications in each area of life [6–9]. The review has put forward a detailed discussion about the classification, different synthetic techniques, biological properties, and applications of AgNPs alongside the biogenic methods used for NP production.

## 2. Classification of NPs

NPs are classified into inorganic, carbon-based, and organic NPs.

### 2.1. Inorganic Based NPs

Those NPs with no carbon atom in their composition, such as metal oxide and metal-based nano-sized particles, are categorized as inorganic NPs.

#### 2.1.1. Metal NPs

Metals such as cadmium (Cd), aluminum (Al), copper (Cu), cobalt (Co), gold (Au), iron (Fe), silver (Ag), zinc (Zn), and lead (Pb) are famously reported in this category. They have unique properties based on their size and characteristics, including expanded surface area, pore size, the density of charge on the surface, cylindrical and spherical shape, color, amorphous, and crystalline structures. Environmental factors, including air, heat, sunlight, and moisture, also affect NP properties (Table 1) [10].

#### 2.1.2. Metal Oxide NPs

As some metals have a higher tendency to make oxides, their oxides are prepared in an attempt to enhance their properties (Table 2). Iron within the presence of oxygen ($O_2$) at room temperature suddenly oxidizes to form iron oxide ($Fe_2O_3$) with improved reactive properties as compared to iron NPs (Table 2). NPs with improved properties, efficiency, and higher reactivity of metal oxide is synthesized like titanic oxide ($TiO_2$), silicon oxide ($SiO_2$), zinc oxide (ZnO), magnetic iron-ore ($FeO_4$), iron oxide ($FeO_3$), aluminum oxide ($Al_2O_3$), and cerium oxide ($CeO_2$). A literature study shows that such NPs have more extraordinary properties than metal NPs [11].

### 2.2. Organic-Based NPs

Organic-based NPs are non-hazardous and eco-friendly, conjointly referred to as nanocapsules (Figure 1). Ferritin, liposomes, micelles, and dendrimers are polymers or organic nano-sized particles with high sensitivity once exposed to light and heat. Because of these distinct characteristics, it is a much better alternative for researchers to prefer organic-based NPs for drug delivery. Its stability, drug-carrying ability, and adsorbing or entrapping drug specificity make them efficient and potential mediators for delivering a drug's active ingredients. These NPs have peculiar surface morphology, size, shape, and composition. Organic NPs are used to inject the active ingredients to the particular site of action within the targeted drug delivery system [12].

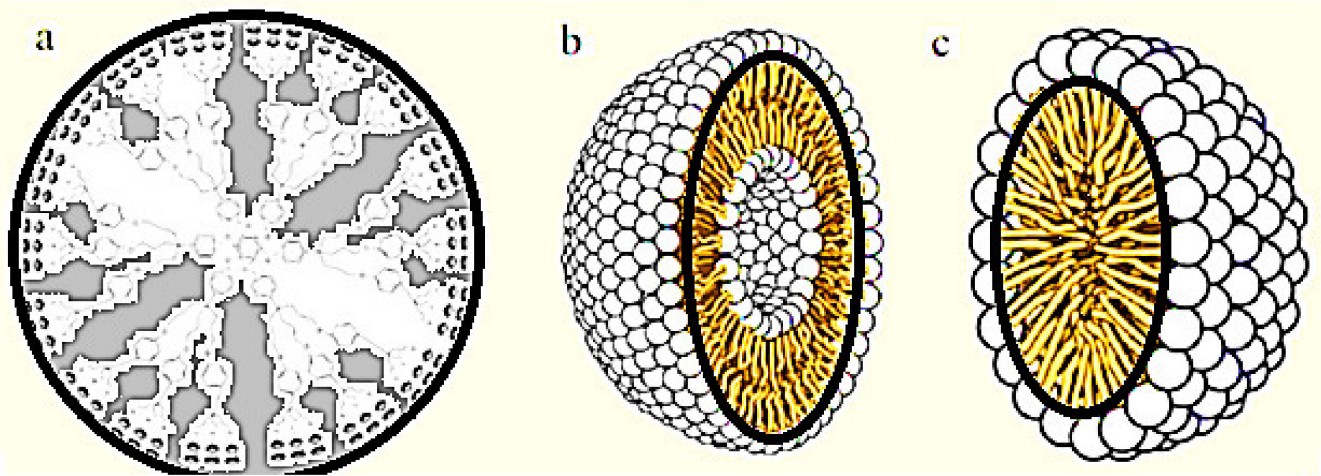

**Figure 1.** Structures of some organic NPs (**a**) dendrimers, (**b**) liposomes, (**c**) micelles.

### 2.3. Carbon-Based NPs

NPs whose skeletons are wholly organized from carbon are referred to as carbon-based NPs (Table 3) [13], which have been categorized into graphene, fullerenes, carbon nanofibers, carbon nanotubes, black carbon, and activated carbon (Figure 2).

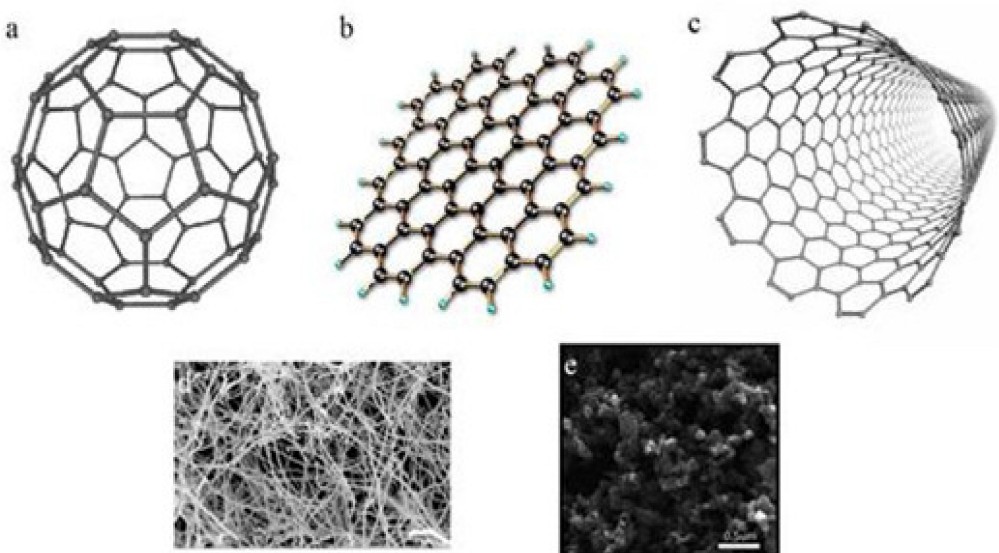

**Figure 2.** Structures of carbon-based NPs (**a**) fullerenes, (**b**) graphene, (**c**) carbon nanotubes, (**d**) carbon nanofibers, and (**e**) carbon black.

### 2.3.1. Fullerenes

Carbon-based NPs could be prepared of carbon atoms bonded through $sp^2$ hybridization having spherical form NP are known as fullerenes. Approximately 28 to 1500 atoms of carbon are combined collectively to form round-shape mono-layered fullerenes with a diameter up to 8.3 nm and poly-layered fullerenes with a diameter of 4–36 nm [14,15].

### 2.3.2. Graphene

Allotropic forms of carbon having a hexagonal structure with two-dimensional planar surfaces are called graphene. A single sheet of graphene has a thickness of 1 nm [13,14].

### 2.3.3. Carbon Nanotubes

Carbon nanotubes are synthesized through coiling the carbons of nano-foil graphene into hollow pipes with a diameter much less than 0.7 nm for monolayer, while for multi-layer, their length modifications range from micrometers to many millimeters, and their ends could either be closed or hollow [14,15].

### 2.3.4. Nanofibers of Carbon

Carbon nanofibers are prepared from nano-foils of graphene by coiling into cup or cone shape instead of regular cylindrical tubes [14,15].

### 2.3.5. Black Carbon

Black carbons are spherical-shaped amorphous materials with a diameter of 20–70 nm. The particles' interactions are much higher, and combine in aggregates when about 500 nm agglomerates are formed [15].

**Table 1.** Classification of metal NPs and their physio-chemical properties.

| NP Type | Reported Physio-Chemical Properties | References |
|---|---|---|
| Zinc NP | Antifungal, antibacterial, anticorrosive, UV filtering | [16] |
| Lead NP | Reactive, high toxicity, highly stable | [17] |
| Copper NP | Highly flammable solids, ductile, very high electrical, thermal conductivity | [18] |
| Cadmium NP | Insoluble, semiconductor of electricity | [19] |
| Gold NP | Reactive, interactive with visible light | [20] |
| Cobalt NP | Magnetic, toxic, absorb microwave, unstable | [21] |
| Silver NP | Disinfectant, antibacterial, absorbs and scatters light, stable | [22] |
| Aluminum NP | Large surface area, highly reactive, sensitive to sunlight, heat, moisture | [23] |
| Iron NP | Sensitive to water and air ($O_2$), reactive, unstable | [24] |

**Table 2.** Classification of metal oxide NPs and their physio-chemical properties.

| NP Type | Reported Physio-Chemical Properties | References |
|---|---|---|
| Aluminum oxide NP | Large surface area, increased reactivity, sensitivity to sunlight, moisture, heat | [25] |
| Zinc oxide NP | UV filtering, antibacterial, anti-corrosive, antifungal | [26] |
| Cerium oxide NP | Low reduction potential, antioxidant activity | [27] |
| Magnetite NP | Highly reactive, magnetic | [28] |
| Silicon dioxide NP | Less toxic, stable, having the ability to functionalize many molecules | [29] |
| Iron oxide NP | Unstable, reactive | [30] |
| Titanium oxide NP | Magnetic character inhibits bacterial growth, high surface area | [31] |

**Table 3.** Classification of carbon-based NPs and their physio-chemical properties.

| NP Type | Reported Physio-Chemical Properties | References |
|---|---|---|
| Fullerenes | Semiconductor, safe and inert, transmits light based on intensity, superconductor, conductor | [32] |
| Graphene | Electrical and thermal conductivity, extreme strength, light absorption | [33] |
| Carbon Nanotubes | Flexible and elastic, high electrical and thermal conductivity, tensile strength | [34] |
| Carbon Nano-fiber | High electrical and thermal frequency, shielding and mechanical properties | [35] |
| Carbon black | Resistant to UV degradation, high strength, electrical conductivity, high surface area | [36] |

## 3. Various Methods Used for the Synthesis of NPS

Generally, the synthesis of NPs is carried out using three different methods given as follows.

### 3.1. Physical Methods of Synthesis

Among physical strategies, evaporation–condensation, spark discharging, and transformation are used for the synthesis of AgNPs. NPs of metals are also synthesized by the thermal breaking technique [37], with an average size within the range of 9.5 nm. Jung et al. [38] reported that metal NPs are prepared using a ceramic ware heater with a close heating area for evaporation of precursor materials. Results indicated that the amount of NPs would increase with the heating of the outside atmosphere. Once there is no flux of the surface heater, the assembly of NPs will increase over time. Even at high concentrations and heat, the spherical-shaped NPs without any agglomeration are determined [39]. AgNPs synthesized within the double deionized water without adding further surfactants is another approach of NPs production within which arc is discharged. The physical synthesis approach for AgNPs typically consumed energies, for instance, arc discharge, arc power, thermal, and so forth. The physical formulation technique is a well-liked method to synthesize AgNPs with massive quantities, although the preliminary cost of the equipment is relatively high with high energy consumption, solvent contamination, and lack of uniform distribution. Various physical methods used for the synthesis of AgNPs are presented in Table 4.

**Table 4.** List of various physical methods used for the synthesis of Ag NPs.

| Method Used | Shape | Size(nm) | References |
|---|---|---|---|
| Laser ablation | Spherical | 31 | [40] |
| Laser ablation | Spherical | 12–29 | [41] |
| Laser ablation | Irregular | 27–41 | [42] |
| Small ceramic heater | Spherical | 6–21.5 | [43] |
| Thermal decomposition | Spherical | 9.5 | [44] |
| Laser ablation | Spherical | 27–120 | [45] |
| Laser ablation | Spherical | 6.48 | [46] |
| Thermal decomposition | Spherical | 14.4 | [47] |
| Laser ablation | Spherical | 4–18 | [48] |
| Laser ablation | Spherical | 5–13 | [49] |
| Laser ablation | Spherical | 20–51 | [50] |
| Thermal decomposition | Spherical | 4–7 | [51,52] |
| Laser ablation | Irregular | 15–20 | [53] |
| Laser ablation | Spherical | 7.9–16.2 | [54] |
| Thermal decomposition | Spherical | 8 ± 1.3 | [55] |
| Thermal decomposition | Spherical | 3.1–4.5 | [56] |
| Thermal decomposition | Spherical | 40–50 | [57] |

### 3.2. Chemical Methods of Synthesis

Chemical methods (Table 5) essentially require metallic precursors, reducing agents, and stabilizing/capping agents. The reduction in silver salts involves nucleation followed by the growth of the NPs. The maximum potently used technique for the synthesis of stable NPs is the chemo-reduction technique. The substances used for AgNPs' synthesis, including elemental hydrogen, ascorbate, citrate and borohydride thio-glycerol, and 2-mercaptoethanol, are toxic.

Besides, due to the potential sedimentation of NPs' surfaces with chemicals, the synthetic NPs are not of predicted purity. It could be challenging to prepare AgNPs with the proper size, requiring a further step to prevent particle aggregation. In addition, during the synthesis procedure, too many poisonous and dangerous byproducts are excised out. The earlier research revealed that solid reductants such as borohydride bring about the formation of tiny particles, which are mono-dispersed to some extent. However, the manufacturing of larger particles may be accomplished by using low reductants such as citrate, having a decreased reduction rate. Thus, the size dispersion can be wide [58]. Chemical strategies employ cytochemical synthesis, laser ablation, lithography, electrochemical reduction, laser irradiation, sonolytic decomposition, thermal decomposition, and chemical reduction. Moreover, the dispersed and unstable prepared metal NPs may be stabilized using specific protective agents. This well-acquainted method is used to shield the NPs with the help of protecting agents that would bind onto or be absorbed in the surface of NPs to prevent accumulation [59].

For instance, the AgNPs may be reduced with the aid of using sodium borohydrate and dodecanethiol for stabilized NPs that prevent their agglomeration in certain solvents [60]. It suggests that a minimal amendment in formulated parameters results in a valuable change in width, size distribution, average size, stability, and shape of the nanomaterial. Generally, polymers such as polymethyl methacrylate, polyvinyl pyrrolidone, polymethacrylic acid, polyethylene glycol, and others are used as stabilizers to manufacture gold NPs [61].

Moreover, a branched polymer methylene-bis-acrylamide aminoethyl piperazine having terminal dimethylamine groups (HPAMAM-N$(CH_3)_2$) has been utilized to obtain colloids of Ag and gold. The piperazine ring, tertiary amine group, and amide impart the branched shape to HPAMAM-N(CH3)2 and are, thus, very critical because of the extra stabilizing and reducing abilities they offer to NPs. NPs of metals may be obtained with peculiar properties such as the low and narrow distribution rates caused by increasing the molar ratio concentration of gold and silver [62]. AgNPs may be synthesized using two phases (aqueous/organic) that rely upon the isolation of reactants in non-mixing solvents. The reaction rate of the reducing agent and precursor of metal is determined by the boundary layer's contact and the inter-phase delivery between the natural and aqueous phases. This response is facilitated by alkylammonium salt of quaternary nature [63]. This method gives us a specific and similar preparation of NPs, through the type of solvent used. The manufacturing of AgNPs may prove very expensive through this method, and also, the produced NPs cannot be isolated from the reaction mixture [64]. The synthesized particles through this method have benefits such as they quickly unfold in natural medium, form stable colloids in non-aqueous media, and catalyze reactions taking place in a non-polar solvent. Their practical applications are gaining tremendous interest, and attempts are being made to put together NPs under diverse chemo-physical environments. However, the solution formed in a non-polar solvent is rare [65]. Therefore, a reduction in Ag and gold is preferably accomplished with the aid of using photoreduction. In aqueous media, the photoreduction in DNA complex metal ions occurs, resulting in metallic solid NPs. The AgNPs organized through this technique have advanced properties if irradiated through UV [66].

**Table 5.** Synthesis of AgNPs using chemical method.

| Silver Salt | Reducing Agent | Capping Agent/Stabilizer | Ag Size (nm) | Reference |
|---|---|---|---|---|
| AgNO$_3$ | Hydrazine hydrate and sodium citrate | Sodium dodecyl sulfate | 10–20 | [67] |
| AgNO$_3$ | D(+)-Glucose and NaOH | – | 8–24 | [68] |
| AgNO$_3$ | Gallic acid | Gallic acid | 7–89 | [69] |
| AgNO$_3$ | Hydrazine hydrate and citrate of sodium | Sodium dodecyl sulfate | 10–20 | [70] |
| AgNO$_3$ | Sodium borohydride | Tri-sodium citrate | 5 | [71] |
| AgNO$_3$ | Aniline | Ethyltrimethylammonium bromide | 10–30 | [72] |
| AgNO$_3$ | Ethylene glycol | Poly vinyl pyrrolidone | 50–175 | [73] |
| AgNO$_3$ | Ethylene glycol | Poly vinyl pyrrolidone | 8–10 | [74] |
| AgNO$_3$ | NaOH | Polyanionic Na + poly($\gamma$-glutamic acid) | 17.2 ± 3.4–37.3 ± 5.5 | [75] |
| AgNO$_3$ | Glucose | Poly vinyl pyrrolidone | 20–80 | [76] |
| AgNO$_3$ | Poly(vinyl pyrrolidone) and gelatin | Glucose, fructose, lactose, and sucrose | 35 | [77] |
| AgNO$_3$ | D-Glucose | Carboxy methyl cellulose, NaOH | 5–15 | [78] |
| AgNO$_3$ | Poly(ethylene glycol) | Poly(ethylene glycol) | 15–30 | [79] |
| AgNO$_3$ | Poly(ethylene glycol) | —- | 10–80 | [80] |
| AgNO$_3$ | Ethylene glycol | Poly(vinyl pyrrolidone) | 17 ± 2 | [81] |
| AgNO$_3$ | Ethylene glycol | —- | 17–70 | [82] |
| AgNO$_3$ | Alkali lignin(low sulfonate) | Alkali lignin(low sulfonate) | 7.3 ± 2.2–14.3 ± 1.8 | [83] |
| AgNO$_3$ | NaOH | Alkali lignin(low sulfonate) | 5–100 | [84] |
| AgNO$_3$ | Sodium borohydride | —- | 3.5–6 | [85] |

*3.3. Biological or Green Methods for the Synthesis of AgNPs*

Biological green synthetic methods are the most convenient alternative to overcome the shortcomings associated with synthetic chemical methods. The greener approach is a simple, value-effective, dependable, and environment-friendly technique for synthesizing AgNPs; ease of solvent medium selection in the type of methods is another advantage (Figure 3). Particular attention has been given to the production of AgNPs of defined size and shape. Diverse biological systems such as plant extracts (Table 6) and microorganisms such as fungi (Table 7), bacteria (Table 8), and small biomolecules such as vitamins and amino acids are used as an alternative source of reducing and capping agents. For instance, the preparation of AgNPs using starch as a coating mediator may act as a reducing agent ($\beta$-D-glucose in a slightly heated assembly) [86].

An aqueous starch solution is used as a solvent to prepare AgNPs to avoid organic solvents. Various types of bacteria such as *Pseudomonas stutzeri*, *Lactobacillus strains*, *Bacillus licheniformis*, *Escherichia coli*, and *Brevibacterium casei*, fungi such as *Fusarium oxysporum* and *Ganoderma neo-japonicum Imazeki*, and plant extracts (*Allophylus cobbe*, *Artemisia princeps*, and *Typha angustifolia)* have been utilized for the synthesis of AgNPs [87]. A reported study shows that a considerable yield of AgNPs may be prepared using an aqueous silver ion (Ag$^+$) solution with *Bacillus licheniformis* used as the stabilizing agent. The fabrication of stable AgNPs with the help of a biogenic approach using microorganisms is advantageous over different techniques because the organisms used are nonpathogenic [88]. Other biomolecules, including biopolymers, starch, fibrinolytic enzyme, and amino acids, are also employed for fabricating AgNPs.

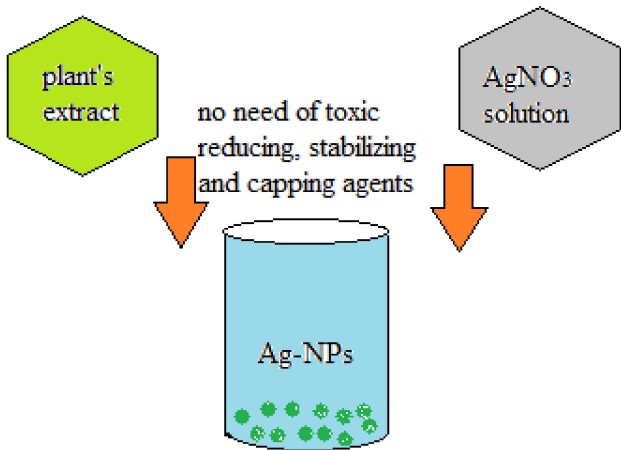

**Figure 3.** Green synthesis of AgNPs.

The green synthesis uses extracts from the different plants as a source of capping and reducing agents in the preparation of AgNPs; it is suggested that the bio-molecules, especially secondary metabolites in those extracts, could reduce the Ag ions. They consist of vitamins, polysaccharides, amino acids, proteins, enzymes, polyphenolics, flavonoids, and other secondary metabolites [89,90]. Likewise, proteins found in the plant extracts also play an essential role as a reductant of the $Ag^+$ and as a size controlling agent for the formulation of AgNPs. Functional groups such as carboxylic (COOH) found in glutamine and aspartic residues and OH group of the tyrosine residues are considered responsible for stabilizing Ag ion and producing small nanoplates of Ag with a little polydispersity [91].

Different plant extracts such as alfalfa, lemongrass soup, geranium leaves, *capsicum annum*, and aqueous geranium extract have been used as bio-reductants to prepare NPs [92–95]. The proteins found in mushrooms can act as a stabilizing agent to reduce Ag ions to obtain NPs of unique shape and a size of approximately 30.6 nm [93]. Glutathione has been extensively utilized as a stabilizing agent to prepare water-soluble AgNPs with many medical side applications [96]. The tryptophan residues present at the C-terminus of the organized oligopeptides are responsible for reducing the Ag ions [96]. According to the Langmuir Blodgett approach, AgNPs may be synthesized using vitamin E as a reducing agent [97–99]. Oleic acid has also been used for the synthesis of biodegradable and green AgNPs [100].

Various microorganisms are employed to expand AgNPs extra-cellularly or intra-cellularly [101,102]. For example, the nanocrystals that contain Ag having altered compositions are prepared using *Pseudomonas stutzeri* bacterium [103]. Fungal species such as *Fusarium oxyporum*, *Chrysosporium phanerochaete*, and *Trichoderma viride* also reduce ($Ag^{+1}$) during AgNPs' synthesis [104]. Furthermore, Ag ion reduction can also be accomplished by *Klebsiella pneumonia*, *Enterobacter cloacae*, and *Escherichia coli* [105].

AgNPs synthesized biologically have characteristic features such as hydrophilic nature, large surface area, and improved stability. This synthesis technique has quite a few benefits as it is economical, simple, regenerable, and needs a small amount of energy compared to particles synthesized by chemical techniques. The AgNP macrophages have been obtained, and their morphology suggests their instability with intermittent rod-like and spherical NPs produced in the diameter extent of 5–40 nm [106,107]. These biosynthesized nano-size particles were studied within the peri-plasmic part of the bacteriological cell among the interior and exterior cellular membranes.

The lactobacillus species have also been used as a source of capping and reducing agents. Lactic acid bacteria can formulate the AgNPs, and both the particle allocation and localization within the cell are dependent on the type of lactobacillus species. Likewise, the thickness of biosynthesized AgNPs produced in this technique varied with the types of lactobacillus. At ambient conditions of temperature and pressure, the reduction in metal



ions with the help of the biological method is much faster than the other techniques [108]. Likewise, Zuorro, A. et al. successfully synthesized AgNPs from bilberry and red currant waste as a source of reducing and stabilizing agents [109].

Similarly, AgNPs were produced in an economical and eco-friendly procedure where phenolic extracts from agro-industrial wastes (bilberry and spent coffee grounds) were used as reducing agents [110]. The biological activity of AgNPs relies upon their morphology and structure managed through the size and shape of the particles. Although AgNPs with unique form and size ranges have been synthesized in different studies, they still have certain shortcomings. A study utilized an extra potent reducing agent such as sodium borohydride (NaBH4) to synthesize uniform-sized silver colloids [108,111]. Though the green synthesis approach has benefits over chemical synthesis, the number of product AgNPs using this approach could be deficient. The shape, size, and distribution of the produced NPs could be optimized by controlling the synthesis techniques, reducing and stabilizing factors, and parameters such as temperature and pH [112].

**Table 6.** Biological method of synthesis of silver NPs using plants extracts as a reducing agent.

| Silver Salt | Plant Origin | Shape | Silver Size (nm) | Reference |
|---|---|---|---|---|
| AgNO$_3$ | *Pinus, Diospyros kaki Ginkgo biloba magnolia and Platanus* | — | 15–500 | [112] |
| AgNO$_3$ | *Artocarpus heterophyllus lam* | Irregular | 10.78 | [113] |
| AgNO$_3$ | *Prunus yedoensis* | Spherical and oval | 20–70 | [114] |
| AgNO$_3$ | *Zingiber officinale* | — | 10–20 | [115] |
| AgNO$_3$ | *Morinda citrifolia* | Spherical | 30–55 | [116] |
| AgNO$_3$ | *Bunium persicum* | Spherical | 20–50 | [117] |
| AgNO$_3$ | *Justicia Adhatoda* | Spherical | 25 | [118] |
| AgNO$_3$ | *Adenium obesum* | Spherical | 10–30 | [119] |
| AgNO$_3$ | *Coffee arabica* | Spherical and ellipsoidal | 20–30 | [120] |
| AgNO$_3$ | *Vigna radiata* | Spherical and oval | 5–30 | [121] |
| AgNO$_3$ | *Jatropha curcas* | Spherical | 10–20 | [121] |
| AgNO$_3$ | *Emblica officinalis* | — | 10–20 | [122] |
| AgNO$_3$ | *Lantana camara* | Spherical | 14–27 | [123] |
| AgNO$_3$ | *Sesuvium portulacastrum L.* | Spherical | 5–20 | [124] |
| AgNO$_3$ | *Mentha peprita* | Spherical | 90 | [125] |
| AgNO$_3$ | *Tribulus terrestris L.* | Spherical | 16–28 | [126] |
| AgNO$_3$ | *Nyctanthes arbor-tristis L.* | Spherical | 50–80 | [127] |
| AgNO$_3$ | *Azadirachta indica* | Spherical | 50–100 | [128] |
| AgNO$_3$ | *Pelargonium sidoides* DC. | Spherical | 16–40 | [129] |
| AgNO$_3$ | *Vigna unguiculata* | Spherical | 24.35 | [130] |
| AgNO$_3$ | *Cinnamomum camphora* | Spherical | 55–80 | [131] |
| AgNO$_3$ | *Aloe barbadensis miller* | Spherical | 15.2 ± 4.2 | [132] |
| AgNO$_3$ | *Amaranthus retroflexus* | Spherical | 10–32 | [133] |

**Table 7.** Synthesis of AgNPs using fungus.

| Silver Salt | Fungus | Shape | Silver Size (nm) | Reference |
|---|---|---|---|---|
| AgNO$_3$ | *Verticillium dahliae Kleb* | — | 25 ± 12 | [134,135] |
| AgNO$_3$ | *Fusarium oxysporum* | Spherical | 5–15 | [136] |
| AgNO$_3$ | *Aspergillus flavus* | — | 8.92 | [137] |
| AgNO$_3$ | *Cryphonectria sp.* | — | 30–70 | [138] |
| AgNO$_3$ | *Pestalotiopsis microspore* | Spherical | 5–25 | [139] |
| AgNO$_3$ | *Phanerochaete chrysosporium* | Pyramidal | 50–200 | [140] |
| AgNO$_3$ | *Cochliobolus lunatus* | Spherical | 3–21 | [141] |
| AgNO$_3$ | *Aspergillus terreus and Penicillium expansum* | Spherical | 6–100 and 14–76 | [142] |
| AgNO$_3$ | *Amylomyces rouxii* | Spherical | 5–27 | [143] |
| AgNO$_3$ | *Aspergillus fumigatus* | Spherical | 17 ± 5.9 | [144] |
| AgNO$_3$ | *Aspergillus niger* | Spherical | 3–30 | [144] |
| AgNO$_3$ | *Alternaria alternate* | Spherical | 20–60 | [145] |
| AgNO$_3$ | *Aspergillus fumigatus* | Spherical | 5–25 | [146] |
| AgNO$_3$ | *Rhizopus stolonifer* | Spherical | 9.47 | [147] |
| AgNO$_3$ | *Cladosporium sphaerospermum* | Spherical | 15.1 ± 1.0 | [148] |

**Table 8.** Synthesis of AgNPs s using bacteria.

| Silver Salt | Microorganisms | Shape | Silver Size (nm) | Reference |
|---|---|---|---|---|
| *AgNO$_3$* | *Bacillus licheniformis* | Spherical | 18–63 | [149] |
| AgNO$_3$ | *Klebsiella pneumonia* | — | 28–122 | [150] |
| AgNO$_3$ | *Pseudomonas antarctica, Pseudomonas proteolytic, Pseudomonasmeridiana* | Spherical | 6–13 | [151] |
| AgNO$_3$ | *Bacillus subtilis* | Spherical | 5–60 | [152] |
| AgNO$_3$ | *Staphylococcus aureus* | — | 160–180 | [153] |
| AgNO$_3$ | *Klebsiella pneumonia* | Spherical | 1–6 | [154,155] |
| AgNO$_3$ | *Nocardiopsis sp. MBRC-1* | Spherical | 45 ± 0.15 | [156] |
| AgNO$_3$ | *Serratia nematodiphila* | Spherical | 10–31 | [157] |
| AgNO$_3$ | *Bacillus subtilis* | Spherical | 20–50 | [158] |
| AgNO$_3$ | *Deinococcus radiodurans* | Spherical | 4–50 | [159] |
| AgNO$_3$ | *Bacillus pumilus* | Spherical | 77–92 | [160] |
| AgNO$_3$ | *Gluconacetobacter xylinus* | Spherical | 40–100 | [161] |

### 3.4. Photochemical Method for AgNP Synthesis

There are two kinds of strategies used for photochemical synthesis: photo-physical (top to bottom) and photochemical (bottom to up). The first technique utilizes metals in their metallic form, and the second one utilizes their ionic precursor. The direct photo-reduction technique is also used for the synthesis of NPs of various metals. Photosensitization is used to prepare AgNPs, wherein metallic ion reduction occurs with excited species generated photochemically [162]. Various light sources such as blue, UV, white, orange, green, and cyan allows the procedure of direct photoreduction in silver nitrate in the presence of sodium citrate at room temperature [163,164]. The specific optical properties related to the size and shape of the NPs are because of the various light sources. Stable AgNPs may

be synthesized via a simple and reproducible UV photoactivation technique in an aqueous Triton X-100 medium [165]. The molecules of Triton X-100 play a double role similar to NPs' stabilizers through capping action and as a reducing agent. Likewise, the dispersion controlled technique wherein the surfactant solution allows the NPs to grow and recover the NPs' size distribution. Another study revealed the successful preparation of AgNPs, while an aqueous alkali solution of silver nitrate was treated with carboxymethylated chitosan under UV light irradiation. The chitosan derivative, carboxymethylated chitosan (CMCTS), is biocompatible, water-soluble, and serves as a reducing agent for $Ag^+$ and the stabilizing agent for NPs of silver in the thickness range of 2–8 nm [165]. The relevant factors of photochemical synthesis techniques are (i) offer photo-induced processing in a disinfected manner, higher accessibility, and better resolution of use; (ii) the satisfactory generation of reducing agent and manufacturing of NPs with the aid of using the photo irradiation; (iii) remarkable flexibility. The photochemical synthesis permits us to formulate the NPs in a unique environment such as emulsion, surfactant micelles, polymer films, cells, and glasses [166].

## 4. Biological Applications of AgNPs

Due to their different properties, AgNPs have been broadly utilized in household utensils, food storage, and health care industry, environmental, and biomedical applications. The present review encompasses a discussion on different biological properties of the AgNPs, emphasizing anti-inflammatory, anticancer, and anti-angiogenic properties and the antimicrobial potential of AgNPs against different classes of microorganism, viz., bacteria, fungi, and viruses.

### 4.1. Antibacterial Potency of AgNPs

AgNPs could help overcome the bacterial resistance against antibiotics in the era of antibiotic resistance by replacing antibiotics as alternative antibacterial agents. A reported research study by Haque et al. [167] validated the antimicrobial activity of AgNPs against *Escherichia coli*, leading to cell death because of the accumulation of AgNPs in the cell wall. The study suggests that the extent of AgNPs' antibacterial activity largely depends on their size and shape [168,169]. In another study, the AgNPs prepared using different saccharides showed potent broad-spectrum bactericidal activity against Gram-positive and Gram-negative bacteria. The noticeable outcome of the study was that the prepared AgNPs were active against multi-resistant bacterial strains such as *Staphylococcus aureus* [169].

Additionally, the efficiency of certain antibiotics could be improved in the presence of AgNPs; for instance, the potential of amoxicillin, penicillin G, and others against *E. coli* and *Staphylococcus aureus* was observed to be increased in the presence of AgNPs in a study. Another study demonstrated the efficiency of AgNPs produced through green synthesis against multiple antibiotic-resistant bacterial strains, including *Staphylococcus aureus*, *Staphylococcus epidermidis*, *Streptococcus pyogenes*, *Klebsiella pneumoniae*, and *Salmonella typhi* [170]. Scientists worldwide are trying to develop new drugs using nanotechnology to decrease antibiotic resistance [171]. Nanotechnology has produced new aspects to treat infections via minute-scale resources and overcome the side effects of available drugs by using green methods [172]. The AgNPs are identified for their excessive antimicrobial/bactericidal activities, and this inhibitory effect can be used to remedy contagious illnesses better [173]. Ag is used as an antimicrobial agent in burns and wounds because microbes can be efficiently eliminated due to the substantial antimicrobial effect of Ag ions. Biofilms are concerned with improving ocular-related infectious illnesses, together with microbial keratitis, which has occurred because of antimicrobial resistance [174]. Abdullah et al. [100] also showed the antimicrobial activity of AgNPs against *Staphylococcus epidermidis* and *Pseudomonas aeruginosa*. Likewise, AgNPs produced using guava leaf extract showed better antibacterial potential against *E. coli* than their chemically synthesized counterpart. The adsorption of bioactive molecules in the extract is considered present at the surface of the NPs and responsible for the more significant antibacterial activity [175].

AgNPs produced by *Cryphonectria* sp. Showed potent antibacterial activity against numerous human pathogenic bacteria, including *S. aureus*, *Salmonella typhi*, *E. coli*, and *Candida albicans*. The mechanism of AgNP-induced cell death (Figure 4), as observed in *E. coli*, is through the leakage of reducing sugars and proteins.

Furthermore, AgNPs can destroy bacterial cell membrane permeability by generating pits and gaps, leading to bacterial cell death [176]. Besinis et al. [177,178] compared the potency of various nanomaterials, including AgNPs, silver, and titanium dioxide of disinfectant chlorhexidine against *Streptococcus mutans*. Among the tested nanomaterials, AgNPs had the most potent antibacterial activity. Agnihotri et al. [179] studied AgNPs' bactericidal action mechanisms using AgNPs immobilized on an amine-functionalized silica surface. They discovered contact killing is the fundamental bactericidal mechanism, and surface-immobilized NPs display more efficacy than colloidal AgNPs. Khurana et al. [180] investigated the antimicrobial properties of silver based on its physical and surface properties against *S. aureus*, *B. megaterium*, *P. vulgaris*, and *S. sonnei*. The enhancement of antibacterial action was observed with particles having a hydrodynamic size of 59 nm compared to 83 nm. The efficacy of the AgNPs relies upon numerous special characteristics such as shape, size, and types of compounds used as reductants and exposure time. These factors have an extensive influence on their bio-medical efficacies [181]. The bio-medical efficacy of AgNPs additionally relies on the sort of microorganisms or plant extract used as stabilizer/reducing agent. As reported earlier, AgNPs showed remarkable antibacterial activities as measured through minimum inhibitory concentration and minimal bactericidal concentration towards the tested bacteria, which is why it is used on a large scale as a bactericidal agent [182].

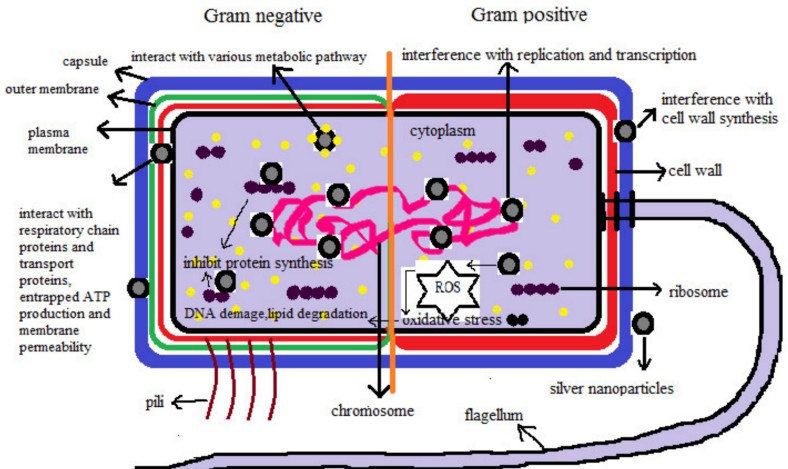

**Figure 4.** Antibacterial mechanism of AgNPs [183].

## 4.2. Antifungal Activity of AgNPs

Fungal infections are commonly present in immunologically suppressed patients. Because of the restricted number of available antifungal drugs, it is troublesome to address all fungal infections. Therefore, it is desperately required to develop biocompatible, non-toxic, and environmentally friendly antifungal agents. AgNPs have been reported to possess significant antifungal activity against several fungal species, including *C. albicans*, *C. tropicolis*, *C. glabrata*, *C. krusei*, *C. parapsilosis*, and *T. mentagrophytes*. The proposed mechanism of action disrupts the fungal envelop arrangement, thereby causing damage to the cells [184]. Similarly, an appreciable antifungal activity of AgNPs was demonstrated by Esteban-Tejeda et al. [185]. Biologically synthesized AgNPs have reportedly potent antifungal activity when combined with fluconazole against *Phoma lomerate*, *Candida glabrata*, *Phoma herbarum*, *Fusarium semitectum*, *Trichoderma sp.*, and *Candida albicans* [183]. Likewise, in another study, antifungal activity of AgNPs has been reported against *Alternaria alternata*, *Macrophomina phaseolina*, *Rhizoctonia solani*, *Sclerotinia sclerotiorum*, *Botrytis cinerea*, and

*Curvularia lunata* [186,187]. Similarly, the AgNPs synthesized by *Bacillus* species displayed increased antifungal activity against the plant pathogenic fungus *Fusarium oxysporum* [188]. Carbon nanoscrolls composed of AgNPs and graphene oxides have shown intense antifungal activity against *Candida albicans* and *Candida tropical*. AgNPs produced through green synthesis in another study strongly suppressed conidial germination and displayed potent antifungal activity against *Bipolaris sorokiniana* [189]. AgNPs also inhibited fungal species such as *Aspergillus fumigatus*, *Penicillium brevicompactum*, *Chaetomium globosum*, *Cladosporium cladosporoides*, *Mortierella alpine*, and *Stachybotrys chartarum* [190].

### 4.3. Antiviral Activity of AgNPs

Viral infections are a common global issue, and it is worsening day by day. Therefore, the development of antiviral agents is a necessary demand of the day. AgNPs can play a vital role due to their antiviral potential and can inhibit the growth and viability of viruses [191–193]. AgNPs showed excellent inhibitory activity against hepatitis B virus (HBV) and human immunodeficiency virus (HIV), as concluded by a reported study [194]. However, the antiviral mechanism of AgNPs remains unknown [195,196]. Lara et al. [197] suggested the primary mechanistic approach on anti-HIV action concerning viral replication taking place at an early stage using polyvinyl pyrrolidone-coated AgNPs that strongly inhibited the transmission of cell-free and cell-associated isolates of HIV-1 [197]. Reported research related to the antiviral action of AgNPs has proven that it can inhibit the viability of viruses; however, the precise mechanism remains unclear [198]. It is also reported that the use of AgNPs considerably prevented the progression of influenza virus in Madin–Darby canine kidney cells (mammalian cell lines) and decreased viral titer in lung tissue [199]. Growth of types 1 and 2 herpes simplex virus and type 3 human parainfluenza virus in the presence of green AgNPs was successfully inhibited [200]. In another study, treatment of AgNPs for 24 h extensively decreased the concentration of virus, prevalence of infection, and severity of bean yellow mosaic virus disease [201].

### 4.4. Anti-Inflammatory Activity of AgNPs

Our body needs a response against the foreign pathogens or particles, which have been triumph over by immunological reactions in the form of elevated levels of pro-inflammatory cytokines, released through the activation of the immune system, and chemotactic materials including complement factors, interleukin-1 (IL-1), TNF-$\alpha$, and TGF-$\beta$ [202–204]. To conquer and stop inflammation for that motive, we require a sufficient amount of anti-inflammatory substances and agents. Among numerous anti-inflammatory agents, AgNPs have performed an essential function as an anti-inflammatory agent. AgNPs are considered an excellent antimicrobial agent; however, their anti-inflammatory responses are still limited. A reported study on the anti-inflammatory action of AgNPs has discovered that it notably reduces the colonic inflammation in rats treated intra-colonially (4 mg/kg) or orally (40 mg/kg) with nanocrystalline silver (NPI 32101) [205,206]. A study carried out by Tian et al. [207] states that in the early stages of wound healing, AgNPs could significantly slow down the production of inflammatory markers by suppressing the anti-inflammatory activities [208]. Biologically synthesized AgNPs can inhibit UV-B-induced cytokine production in HaCaT cells and reduce paw tissues' edema and cytokine levels [209].

### 4.5. Anti-Angiogenic Activity of AgNPs

Inflammatory and ischemic diseases are caused due to pathological angiogenesis and certain types of cancers [210]. Studies are on the way to seek anti-angiogenic substances to address angiogenic ailments better. Many synthetic anti-angiogenic substances are available, but natural pro- and anti-angiogenic elements that would contribute an effective physiological strategy for treating every type of antigenic ailments are still needed. Recently reported in vitro and in vivo studies have suggested their anticancer and anti-angiogenic properties [211]. Chan et al. [212] have recommended using biologically synthesized AgNPs in the bovine retinal endothelial cells (BRECs) model and shown significant inhibition

of proliferation and migration in BRECs. Their results revealed that AgNPs strongly inhibited the proliferation and migration in BRECs after 24 h treatment and exhibited anti-angiogenic properties. The mechanism of inhibiting vascular endothelial growth factor (VEGF) introduced an angiogenic process via caspase-3 and DNA fragmentation activation. Reported studies show that AgNPs strongly inhibit the VEGF-induced PI3K/Akt pathway in BRECs [213]. Tavakolpour and Karami [214] also reported the anti-angiogenic property of AgNPs. AgNPs inhibited VEGF-induced angiogenic activity via blocking the formation of new blood microvessels and inactivation of PI3K/Akt in BRECs. Fibroblast growth factor-induced angiogenesis can also be inhibited using AgNPs [215].

*4.6. Anticancer Activity of AgNPs*

Cancer is a major life-threatening illness, and one out of three individuals can develop cancer in their lifetime. Although for the therapy of various types of cancer, different chemotherapeutical agents are presently offered with numerous side effects, and specifically, the administration of chemotherapeutic agents intravenously is an insufficient method [216]. Consequently, it is necessary and a point of interest for researchers to formulate nanomaterial with no side effects and target the desired cells or tissues with accuracy. The anticancer activities both in in vitro and in vivo model systems have been reported [216]. Quan et al., in their study, demonstrated that cancer cell death is dependent on the concentration of AgNPs [217].

Moreover, synergistic results on cell death using uracil phosphoribosyl enzyme (UPRT)-expressing cells and non-UPRT-expressing cells in the presence of fluorouracil and the induction of apoptosis and sensitization of cancer cells by AgNPs were studied. They showed anticancer properties once coated with starch as studied in healthy human lungs fibroblast cells (IMR-90) and glioblastoma cells (U251). Changes in cell morphology, reduced metabolic activity, cell viability, and accumulated reactive oxygen species (ROS) were induced by AgNPs [218].

Song et al. [219] prepared multifunctional silver-embedded magnetic NPs; the silica-encapsulated magnetic NPs produce surface-enhanced solid Raman scattering signals to target breast cancer cells and floating leukemia cells [220]. A study showed developed carrier molecules such as chitosan to deliver silver to the cancer cells rather than using AgNPs directly. Likewise, Ullah et al. demonstrated that at low concentrations, chitosan-based nano-carrier delivery of AgNPs could induce apoptosis [221]. Similarly, two individual studies have shown that chitosan-coated silver nano-triangles improve the cell mortality rate [222] and possess significant cytotoxic activity in acute myeloid leukemia cells [223]. AgNPs using bacterial and fungal extract demonstrated significant cytotoxicity in human carcinoma cells [224,225].

Plant extract-mediated synthesis of AgNPs has shown selective cell-specific toxicity of cancer cells in human lungs [226]. Another reported study reveals that targeted delivery could be a crucial method for cancer treatment. To deal with this issue, Locatelli et al. [227] developed multifunctional nanocomposites comprising compound NPs (PNPs) and AgNPs. PNPs conjugated with a chlorotoxin showed a non-linear dose–effect relationship, whereas the toxicity of chlorotoxin remained stable. Biologically synthesized AgNPs exhibited vital toxicity in cancer cells (MCF7 and T47D) by higher endocytic activity than the regular breast cell line (MCF10-A) [228].

## 5. Water Purification and Treatment

Purification of drinking water is a necessity of the era, as the water from different sources may potentially be a source of toxic levels of microorganisms, heavy metals, and organic molecules [229]. Strategies such as coagulation, filtration, settlement, chlorination, and many different chemical approaches purify water for consumption in the household [230]. AgNPs with higher stability, cost-effectiveness, and with a controllable release rate have been successfully employed for the removal of inorganic anions [231], heavy metals [232], organic pollutants [233], and bacteria [234] from water and have shown promising

potential for application in water and wastewater treatment. In recent years, their use for water disinfection has especially increased because of their high toxicity to microorganisms. However, direct application of AgNPs could lead to their aggregation in aqueous media leading to a gradual reduction in their efficiency in long-term use [235]. In this context, AgNPs attached to filter materials could be a better alternative to minimize the aggregation problem and be cost-effective with efficient antibacterial potential, as exhibited in different studies [236]. The sheets where AgNPs are deposited on the cellulose fibers have shown significant antibacterial properties against *E. coli*. Moreover, Ag$^+$'s loss from such sheets does not exceed its standard range in drinking water (0.1 ppm) set by the environmental protection agency and the WHO [237].

Moreover, the effective usage of Ag NPs embedded on ceramic materials/membranes for the disinfection and treatment of water for household use at the point of use has also increased in the last two decades [238]. Yet another use of AgNPs in water treatment is to prevent fouling of the membrane filters used in water treatment systems [239,240].

*5.1. The Catalytic Efficiency of AgNPs for Water Pollution Monitoring Using Different Treatment Methods*

The catalytic application of NPs for water purification may be summarized for three fundamental varieties of contaminants: halogenated organics that include pesticides, heavy metals, and microorganisms [241]. By changing the characteristics of surface functionalization of AgNPs and selecting specific ligands, the particles can be made selective to specific analytes and increase their sensitivity. In their study, Que, Z. G. et al. reported the successful synthesis of AgNPs supported on zirconia–ceria to improve the Ag dispersion on the catalyst support and inhibit its sintering during the reaction and used the synthesized NPs for the catalytic wet air oxidation of refractory organic compounds [242]. Of the various types of wastewater, printing and dye wastewater represent the most difficult problem due to their complex components [243]. The large amounts of non-degradable oil and toxic 4-nitrophenol dissolved in wastewater pose an urgent challenge [244]. AgNPs have a role in the catalytic degradation of 4-nitrophenol [245] due to the high specific surface, abundantly exposed low-coordination sites, and low cost. However, due to the surface's high energy, there is a tendency for decreased catalytic activity over time. In addition, the powders are very difficult to recover from the water, which leads to possible secondary contamination. Therefore, immobilization of AgNP in porous oil absorbent materials has been observed as an effective approach to improving the recyclability of AgNP [246].

In another study, multifunctional 3D filter cotton was successfully manufactured by simply immobilizing PDA and AgNP on the surface. The high catalytic degradation performance of the processed 3D filter cotton can be used to degrade water-soluble 4-nitrophenol and has excellent oil/water separation efficiency and recyclability [247]. Likewise, green synthesis of highly stable AgNPs and their effective role as catalyst, photocatalyst, and antimicrobial agent has been reported in a study to treat wastewater [248].

The increasing integration of AgNPs in environmental applications, including controlling and treating water pollution, raises concern about their effects on the environment. Grafting AgNP onto specifically selected polymers represents a potential solution to overcoming environmental safety concerns and the coalescence of nanoparticles. AgNP–cellulosic hybrids have the dual advantages of being easily made with recycled material, having low cost and potential reuse, and being environmentally friendly when properly designed to control and treat water pollution [249].

Manivannan et al. [250] developed AgNPs embedded in an amine-functionalized silicate sol-gel matrix, produced using a different combination of silicate, surfactant, and cyclodextrin. The selective detection of Hg (II) ions by the AgNP-based sensors in 500 mM environmentally relevant metal ions was verified using spectral and colorimetric methods. Another important characteristic of AgNPs is their potential as photocatalysts. Sharma et al. [251] investigated the same behavior for their AgNPs. They developed a simple, label-free, inexpensive, portable, selective, and sensitive colorimetric sensor based on thiol-modified chitosan AgNPs for the real-time detection of toxic Hg (II) ions in water.

Melinte et al. [252] prepared several photocatalysts primarily based on Ag, Au, or Au-Ag nanoparticles supported on photo cross-linked natural, and those hybrid structures allowed the photocatalytic degradation of polymers 4-nitroaniline. Roy et al. [253] studied the photocatalytic degradation of methylene blue dye in the presence of biogenic AgNPs synthesized using yeast (*Saccharomyces cerevisiae*) extract. To sum up, several documented reports on the catalytic degradation efficacy of AgNPs show the capability of their potential use in water treatment systems. However, while designing a water treatment system based on AgNPs, their potential release and possible secondary water contamination should be considered.

*5.2. Role of the Antibacterial and Antifungal Potency of AgNPs in Water Treatment*

The documented antimicrobial potency of AgNPs makes them perfect candidates for their use in water treatment and disinfection. The antimicrobial activity of AgNPs as proposed in the literature is because of any of these three actions: (1) alteration of membrane properties; (2) damage to DNA/RNA or proteins; (3) release of Ag (I) in the cell cytoplasm. The antimicrobial potential of AgNPs is why several AgNP-based products are used in disinfecting water and as water filters to avoid pathogenic bacteria and viruses. However, it was reported [254] that applying AgNPs directly to water might result in their bioaccumulation in fish cultures. Therefore, toxicity studies might be required to determine aquaculture uses. Another study [255] has evaluated equipment-covered filters with AgNPs to reduce fungal growth in aquacultures; the use of AgNPs in filters was found significantly effective in preventing fungal infections without observing side effects. The antimicrobial activity of AgNPs is closely related to the release of Ag+ ions in the aquatic environment [255].

Similarly, Deshmukh, S.P et al. developed AgNP-based biofilters for disinfection of water by eliminating ammonia and other contaminants, including microorganisms from water [256]. In another study [256], nanotubes were elaborated with carbon, and cyclodextrin nanotubes impregnated with AgNPs were prepared to purify water samples in which *E. coli and V. cholerae* were detected. The reported studies from the literature safely conclude the potential use of AgNPs as water disinfectants for three major reasons: (a) its exceedingly high antimicrobial activity, (b) stability and slow-release, and (c) potential of large-scale production through green synthesis.

## 6. Potential Hazardous Effects of AgNPs

As we acknowledge the diverse applications of AgNPs and their composites, they have some negative impacts. AgNPs are utilized in digital devices, toys, cleansing, scientific appliances, and the food industry [257]. They could prove a potential threat of toxicity to aquatic organisms, and the extensive use of AgNPs as antimicrobial agents and disinfectants could enhance bacterial resistance [258]. The maximum limit of AgNPs' additives in drinking water is 100 μg/l; however, it is expected that the dissolved silver/nanosilver levels rise to toxic levels because of the release of nano-size silver from AgNPs. The AgNP-caused toxicity is higher for aquatic species than terrestrial animals and humans [259]. Environmental toxicity is associated with the consecutive release of nano-size silver, and their effect on marine life is because of the distribution and emission of the products [260]. The toxicity of the AgNPs, as described by a literature review, is due to surface phenomena such as aggregation, sulfidation, and phase transformation in aquatic organisms. It also reveals silver toxicity for some terrestrial, aquatic plants, fungi, algae, human cells, pores, and skin, and other vertebrates [261]. Another study states that AgNPs, because of the accumulation of particles in the cell medium, cellular uptakes, localization inside the cell, and its discharge and release in the lung cells, are responsible for the toxicity of AgNPs [262]. Concluding studies about the determination of the toxicity limit concentration of AgNPs are inadequate as the extent of their potential toxicity depends on many factors, including the concentration of AgNPs and their size, shape, and surface area. Moreover, their sources, route of entry to the body, methods of

toxicological assessment (in vitro, in vivo, in silico), and dose units (ppm, mass per volume, or mass per unit of NPs) may significantly alter and, hence, make it difficult to define the exact toxicity range of AgNPs. Nonetheless, AgNPs have more potential toxicity compared to $Ag^+$ ions [263].

### 7. Future Perspectives and Conclusions

This review article summarizes the classification, strategies of synthesis, and applications of AgNPs in depth. The review offers a ready reference source in a single place for a detailed discussion about different characteristics and types, synthesis strategies, biological properties, medicinal applications, the role in water treatment and purification of AgNPs, and the potential hazardous effects associated with them. Different strategies such as physical, chemical, and biological approaches widely used for AgNPs' synthesis have been discussed in detail. As described and stated in the article, the preferred synthesis methodology is green synthesis because of its environmental compatibility, low cost, simplicity of the process, re-generation, and need for a small amount of energy. In nano-chemistry, particle size is the most critical factor, and it has been observed that using different plants, different size particles of the same metal could be obtained, clearly indicating that different plants have different capabilities of reducing the metal ions. However, the sole factor ascertained here is that the NPs' quantity is lower through green synthesis. There ought to be a side effects assessment, which is also a problem that requires intellectual attention.

AgNPs, because of their considerable antimicrobial potential, are employed for the treatment of wastewater and household water. The review covers the different strategies of AgNPs' use for water purification and disinfection. The documented biological potentials of AgNPs discussed in the review encourage scientists to utilize different plants as a source of reducing agents to obtain the required particle size and shape and with varying medicinal applications.

**Author Contributions:** N.N.; Conceptualization, N.N.; methodology and writing—original draft preparation, M.I. and S.N.; software, S.N.; validation, I.Z.; formal analysis, A.K., F.U. and J.B.; investigation, A.K. and A.W.K.; resources, M.Z.; data curation, N.P.; writing—original draft preparation, I.Z. and N.N.; writing—review and editing, M.Z. and F.A.K.; visualization, I.Z.; supervision, M.Z.; project administration, M.Z., A.K. and N.P.; funding acquisition, M.Z. All authors have read and agreed to the published version of the manuscript.

**Funding:** This research was funded by project number T190087MIMV and the European Commission, MLTKT19481R "Identifying best available technologies for decentralized wastewater treatment and resource recovery for India" and SLTKT20427 "Sewage sludge treatment from heavy metals, emerging pollutants and recovery of metals by fungi," and by projects KIK 15392 and 15401 by the European Commission.

**Institutional Review Board Statement:** Not applicable.

**Informed Consent Statement:** Not applicable.

**Data Availability Statement:** All the data is presented in this paper. No data is there in any other repository.

**Conflicts of Interest:** The authors declare no conflict of interest. The funders had no role in the study's design, collection, analyses, and interpretation of data, manuscript writing, or decision to publish the results.

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
