# Peer review of "A Review on Silver Nanoparticles: Classification, Various Methods of Synthesis, and Their Potential Roles in Biomedical Applications and Water Treatment"

_water, doi:10.3390/w13162216_

Round 1
Reviewer 1 Report
Dear Authors,
I read your paper and it is mainly the same as at the initial submission. Few pages have been added but until that part it is not obvious why the article was submitted to Water. Moreover the novelty of your study is not clear. Critical comparisons of the methods should de frequent.
On my opinion the article looks like an introductory course for students or a book chapter.
Therefore,I cannot recommend the publication.
Author Response
Reviewer 1
I read your paper and it is mainly the same as at the initial submission. Few pages have been added but until that part it is not obvious why the article was submitted to Water. Moreover the novelty of your study is not clear. Critical comparisons of the methods should de frequent.
On my opinion the article looks like an introductory course for students or a book chapter.
Therefore, I cannot recommend the publication.
- Dear reviewer, the paper has been revised thoroughly. Extra details have been omitted. The review has now shortened to a few specific topics, including introducing silver nanoparticles, their types, biological and medicinal applications, their role in water treatment, and possible toxicity. Special efforts have been put in this time to make the review concise and to the point.
- While revising, special attention has been given to the uses of AgNPs in water treatment and purification. New insights, information, and a comparison of the different methods from the literature have been added (highlighted for your attention, please) to describe the relationship of AgNPs with water treatment.
Reviewer 2 Report
To my point of view the manuscript water-1325816 did not undergo the expected modifications. The manuscript is potentially publishable, but it is still weak and must be further improved before publication.
Specific comments:
- The added sections "7. Catalytic efficiency of AgNPs for Water Pollution Monitoring using different Treatment methods" and "8. Antibacterial and antifungal potency of Silver nanoparticles to improve water quality" seem to be hastily and roughly written with many typos etc. Please re-write.
- English needs to be checked throughout the text.
- Check the reference numbering to the whole document. There are many discrepancies.

Author Response
Reviewer 2
To my point of view the manuscript water-1325816 did not undergo the expected modifications. The manuscript is potentially publishable, but it is still weak and must be further improved before publication.
Specific comments:
- The added sections "7. Catalytic efficiency of AgNPs for Water Pollution Monitoring using different Treatment methods" and "8. Antibacterial and antifungal potency of Silver nanoparticles to improve water quality" seem to be hastily and roughly written with many typos etc. Please re-write.
- Dear reviewer, thank you for the recommendations. The article has been thoroughly revised. The section mentioned by your worthy self has been re-written, and corrections have been made as suggested.
- English needs to be checked throughout the text.
- It is updated accordingly.
- Check the reference numbering to the whole document. There are many discrepancies.
- It is revised as suggested.
Reviewer 3 Report
Muhammad Zahoor et al. have presented the review paper on silver nanoparticles, their properties, and biomedical applications.
A revised version of the paper is now well-organized. All recommended changes have been addressed by the authors. Newly added text and references are perfect.
I now recommend that this paper can be accepted in Water in its present form.
Author Response
Reviewer 3:
Muhammad Zahoor et al. have presented the review paper on silver nanoparticles, their properties, and biomedical applications.
A revised version of the paper is now well-organized. All recommended changes have been addressed by the authors. Newly added text and references are perfect.
I now recommend that this paper can be accepted in Water in its present form.
- Thank you worthy reviewer for your positive input.
Round 2
Reviewer 1 Report
The comments sent by me in the first review were taken into account by
the authors and the manuscript was corrected.
Best regards,
Reviewer 2 Report
The suggested modifications were made. I suggest publication in present form.
This manuscript is a resubmission of an earlier submission. The following is a list of the peer review reports and author responses from that submission.
Round 1
Reviewer 1 Report
A Review paper by Muhammad Zahoor et al. on the synthesis and role of silver nanoparticles for various biomedical applications highlights a brief overview of various synthesis techniques and their uses. Considering the growing demands of nanoscale silver due to its efficiency for various applications, the authors have nicely summarized it in a single paper. Authors have classified various nanomaterials in three categories based on their chemical nature, i.e., organic, inorganic, and carbon-based. Detailed classification of the carbon-based techniques seems out of the interest of this review, but the readers from completely different fields can get the benefit of reading all types of nanoparticles in one place.
Overall, the review paper looks good. Though authors claimed that the silver nanoparticles can be used in almost all possible biomedical-related applications. This is more towards perspective rather than a review.
-Since most applications of silver nanoparticles shown are biomedical related, the title of the paper needs to be revised. It should be something like:
A Review on Silver Nanoparticles: Classification, Various Methods of Synthesis, and Their Roles in Potential Biomedical and other Applications
-There are lots of spelling mistakes in texts, authors need to address those during revision.
-There are lots of redundancies of the word, especially on page 19: such as flexibility, flexibility (Line 3, page 19), fitness and fitness (second paragraph, line 2, page 19), is due to is due to (second paragraph), etc.
-Also, some words like not U/V but UV, not AGNP but AgNP on page 19 needs to be corrected.
Overall, the review paper is good and I am recommending it for publication in Water after thorough revision in terms of spellings, removing redundancies, etc. Also, authors may consider removing some unnecessary applications description in texts.
Reviewer 2 Report
This manuscript water-1264615 reviewed the current knowledge of various methods of AgNPs synthesis, the biological activities of AgNPs, their role in water purification and other potential applications. 278 published papers were included in this review. The overall quality of the work is high, and the paper well written. Thus, it can be accepted after minor revision.
Specific comments and suggestions are provided below to help the authors develop and improved manuscript.
- Authors should check and correct the numbering of paragraphs to the entire text. For example:
Page 18, line 617. ʺ1.1.1. Water purificationʺ should be replaced with ʺ2.7.1. Water purificationʺ.
Page 19, line 663. Replace ʺ1. Side Effects or Hazardous Effects of AgNPSʺ with ʺ3. Side Effects or Hazardous Effects of AgNPsʺ.
- Page 18, line 624. Replace ʺ[261-163]ʺ with ʺ[261-263]ʺ.
- Page 19, line 681. Replace ʺAGNPsʺ with ʺAgNPsʺ
- Page 19, line 684 and line 695. Sections ʺ2. Future Perspectives and Conclusionsʺ and ʺ3. Conclusionʺ should be combined in one section: ʺ4. Future Perspectives and Conclusionsʺ that needs to be re-written and expanded to clearly state future research needs. Try to avoid any repetitions.

Reviewer 3 Report
Dear Authors,
The paper entitled “A Review on Silver nanoparticles: Classification, Various Methods of Synthesis, Their Role in Water Purification, and Other Potential Applications” present an interesting integrative framework project for nanoparticles, but I consider that all information is, in general, only as in an encyclopedia and isn’t at all correlated with the topic of the special issue - Efficient Catalytic and Microbial Treatment of Water Pollutants.
I recommend taking into consideration the following:
Point 1 – Please, present the results and also schematic organizer data of different studies (yours and other researches) about the catalytic efficiency of AgNPs in the water’s treatment (surface or groundwaters and wastewaters);
Point 2 – Please, correlate the antibacterial potency and antifungal activity of AgNPs with water treatment;
Point 3 – Please, add the critical comparative analysis of different water treatment methods by using AgNPs.
Point 4 - Please, explain what be the relationships between Hospital Acquired Infections, Diagnostic, Biosensor, and gene medical care Applications of AgNPs, Targeted Drug Delivery, Wound Healing or Anticancer Activity of AgNPs with Water Treatment?
and some corrections in text or graphic form:
R111-112 - please, fit the dimension of Figure 1 if it is possible at the same size as Figure 2.
R147-158 – why is necessary the abbreviation terms? The characterization techniques of NPs aren’t present in the manuscript.
Please, be careful with the numbering of paragraphs or sections; e.g.
R617 – 1.1.1. Water purification;
R663 – 1. Side Effects or Hazardous Effects of AgNPS ;
R684 – 2. Future Perspectives and Conclusions;
R695 – 3. Conclusions
Also, please be much more transparent in the phrase (R691): There ought to be a side effects assessment, which is also a problem that required intellectual attention.